# Mitogenomic Insights into the Evolution, Divergence Time, and Ancestral Ranges of *Coturnix* Quails

**DOI:** 10.3390/genes15060742

**Published:** 2024-06-05

**Authors:** Prateek Dey, Swapna Devi Ray, Venkata Hanumat Sastry Kochiganti, Budhan S. Pukazhenthi, Klaus-Peter Koepfli, Ram Pratap Singh

**Affiliations:** 1Sálim Ali Centre for Ornithology and Natural History (South India Centre of Wildlife Institute of India), Anaikatti, Coimbatore 641108, Tamil Nadu, India; pratikdey23@gmail.com (P.D.); swapnadray555@gmail.com (S.D.R.); 2Bharathiar University, Coimbatore 641046, Tamil Nadu, India; 3Center for Species Survival, Smithsonian Conservation Biology Institute, National Zoological Park, Front Royal, VA 22630, USA; pukazhenthib@si.edu; 4National Institute of Animal Nutrition and Physiology, Bengaluru 560030, Karnataka, India; kvhsastry@hotmail.com; 5Smithsonian-Mason School of Conservation, George Mason University, Front Royal, VA 22630, USA; 6Department of Life Science, Central University of South Bihar, Gaya 824236, Bihar, India

**Keywords:** *Coturnix*, ancestral ranges, positive selection, dispersal

## Abstract

The Old-World quails, *Coturnix coturnix* (common quail) and *Coturnix japonica* (Japanese quail), are morphologically similar yet occupy distinct geographic ranges. This study aimed to elucidate their evolutionary trajectory and ancestral distribution patterns through a thorough analysis of their mitochondrial genomes. Mitogenomic analysis revealed high structural conservation, identical translational mechanisms, and similar evolutionary pressures in both species. Selection analysis revealed significant evidence of positive selection across the *Coturnix* lineage for the *nad4* gene tree owing to environmental changes and acclimatization requirements during its evolutionary history. Divergence time estimations imply that diversification among *Coturnix* species occurred in the mid-Miocene (13.89 Ma), and their current distributions were primarily shaped by dispersal rather than global vicariance events. Phylogenetic analysis indicates a close relationship between *C. coturnix* and *C. japonica*, with divergence estimated at 2.25 Ma during the Pleistocene epoch. Ancestral range reconstructions indicate that the ancestors of the *Coturnix* clade were distributed over the Oriental region. *C. coturnix* subsequently dispersed to Eurasia and Africa, and *C. japonica* to eastern Asia. We hypothesize that the current geographic distributions of *C. coturnix* and *C. japonica* result from their unique dispersal strategies, developed to evade interspecific territoriality and influenced by the Tibetan Plateau’s geographic constraints. This study advances our understanding of the biogeographic and evolutionary processes leading to the diversification of *C. coturnix* and *C. japonica*, laying important groundwork for further research on this genus.

## 1. Introduction

The genus *Coturnix*, representing nomadic Old World quails, is widespread across the grasslands of Africa, Europe, Asia, and Australia [1]. As the sole long-distance migratory members of the Galliformes family Phasianidae, these quails offer unique insights into avian dispersal and adaptation [1,2]. Amongst the six described species, of particular interest are *C. coturnix* (common quail) and *C. japonica* (Japanese quail), two species that are morphologically indistinguishable in field conditions, with vocalizations often serving as the primary means of identification [1,3,4]. Despite their similar appearance and overlapping life history traits, these species exhibit distinct geographic distributions, with limited sympatry in regions such as Mongolia and northeastern India [1,3,4].

The behavioral and morphological similarities between *C. coturnix* and *C. japonica* are expected to extend to the molecular and genetic levels, yet this assumption has not been empirically tested. Furthermore, it would be scientifically intriguing to explore if the seemingly demarcated distribution of *C. coturnix* and *C. japonica* can be understood by an assessment of their phylogenetic topology and evolutionary history. To address this, our study utilizes mitogenomes to analyze the evolutionary trajectory and biogeographic patterns of *C. coturnix* and *C. japonica*. Mitogenomes are characterized by high mutation rates and low recombination, making them an ideal marker to study recently diversified species and their phylogeographic patterns [5,6].

Although previous studies have shed light on various aspects of *Coturnix* phylogeny and evolution, there remain unanswered questions regarding the divergence and biogeography of *C. coturnix* and *C. japonica*. For example, while Kimball et al. analyzed the whole mitogenomes of both species, they did not provide divergence time estimates or biogeographic reconstructions [7,8]. Chen et al. estimated divergence times for Galliformes using genomic ultraconserved elements but did not include *C. coturnix* in their sampling [9]. Stein et al. examined divergence estimates using mitochondrial genes and nuclear loci for Galliformes, encompassing *C. coturnix* and *C. japonica*, yet omitted biogeographic reconstructions [10]. Moreover, Wang et al. conducted an extensive study on divergence estimates and ancestral ranges within Galliformes, but their analysis only included *C. coturnix* [11]. Therefore, no study has exclusively analyzed the complete mitogenomes of *C. coturnix* and *C. japonica* to estimate their divergence dates, reconstruct their ancestral ranges, or assess their evolutionary trajectories in a comparative framework. Additionally, mitogenome sequences of *C. coturnix* and *C. japonica* from the Indian subcontinent have not been documented, and a comprehensive characterization of these mitogenomes, including structural features, codon usage patterns, and evolutionary attributes, is lacking.

We hypothesize that the morphological similarities between these species will be mirrored in their mitochondrial genome structures, molecular mechanisms, selection pressures, and phylogenetic relationships. Furthermore, we anticipate that differences in their ancestral ranges have led to distinct dispersal routes, culminating in their present distribution. Therefore, this study aims to (i) comprehensively characterize mitochondrial genome data for *C. coturnix* and *C. japonica*, (ii) investigate mitochondrial genome-based divergence estimates and range reconstructions for these species, extending this to other *Coturnix* quails, and (iii) explain the evolutionary trajectory and current geographic distributions of *C. coturnix* and *C. japonica*.

## 2. Materials and Methods

### 2.1. Sample Collection

A single specimen of *C. coturnix* was collected through a road-killed survey in Nagpur, Maharashtra State, India, with due permission from the Maharashtra Forest Department (Desk-22(8)/Research/CR-8(18-19)/875/2018-19). Identification of the specimen was performed in the field using a field guide [12]. Muscle, liver, heart, and testes (identifying the sample as male) tissues from the carcass were sampled and stored in DESS buffer (20% DMSO, 0.25 M tetra-sodium EDTA, sodium chloride till saturation, pH 7.5). The approximate time of tissue collection was estimated to be within 2 h of the road-hit occurrence. Under suitable conditions (0 °C, sterile ice box), the tissues were transported to the National Avian Forensic Laboratory at the Sálim Ali Centre for Ornithology and Natural History (SACON), Coimbatore, Tamil Nadu State, India, and stored at −80 °C until DNA extraction. Using a tissue lysis buffer (10 mM Tris-pH 8.0, 10 mM EDTA-pH 8.0, and 100 mM NaCl; 0.35 mM SDS and 34 units Proteinase K), ~20 mg of muscle tissue was digested at 52 °C for 12–15 h in a heating block. DNA extraction was performed using a modified phenol, chloroform, and isoamyl alcohol method and was stored under voucher code NAFL/0322/DNA/080720 [13]. The quality of the extractions was assessed using 1% agarose gel electrophoresis and quantified using spectrophotometry (DeNovix, Wilmington, DE, USA) and a Qubit 4 Fluorometer using the dsDNA High Sensitivity Assay Kit (ThermoFisher Scientific, Waltham, MA, USA).

For *C. japonica*, a commercially farmed specimen from India was sampled from the market, and the biomaterials (muscle tissue and blood) were stored in DESS buffer and transported to the lab under suitable conditions as stated above. The specifics of the *C. japonica* sample and the whole genome sequences produced are detailed in a prior publication by our research group [14]. To construct the *C. japonica* mitochondrial genome, we utilized the raw sequencing reads as reported in [14].

### 2.2. Library Preparation, NGS Sequencing, and Mitogenome Assemblage for C. coturnix 

The wet-lab and next-generation sequencing methods used in this study were carried out and developed in-house as described below [15,16,17,18]. Extracted DNA was utilized for library preparation following the TruSeq DNA PCR-Free library preparation kit protocol (Illumina Inc., San Diego, CA, USA). About 1100 nanograms of the isolated genomic DNA were used as starting material to generate a paired-end genomic library of insert size 350 (2 × 150) bp. A focused ultrasonicator (Covaris M220, Woburn, MA, USA) was used to fragment the genomic DNA to the desired fragment size. Subsequently, TruSeq kit reagents were used to clean up the fragmented DNA, create blunt ends, and ligate adapters to the library fragments. The mean peak size of the library fragments was assessed using a Fragment Analyzer AATI 5200 (Agilent, Santa Clara, CA, USA) and quantified using the QIAseq Library Quant Assay Kit (Qiagen N.V., Hilden, Germany). The library was sequenced using the NextSeq550 instrument (Illumina Inc., USA).

bcl2fastq v2.20.0 software (Illumina Inc., USA) was used for sample de-multiplexing and adapter trimming. Overall quality was assessed using FastQC (https://www.bioinformatics.babraham.ac.uk/projects/fastqc/, accessed on 1 December 2020). Raw reads of *C. coturnix* and *C. japonica* were mapped to a reference mitogenome of *C. japonica* (NCBI acc. no. NC003408.1; [19]) using the Geneious mapper embedded in Geneious Prime version 2023.2.1 (https://www.geneious.com). Medium sensitivity, 25 iterations, and the highest quality threshold options were used in Geneious Prime to successfully assemble the *C. coturnix* and *C. japonica* mitogenomes from raw reads. The prediction of structural features in the newly obtained mitogenomes was carried out using the MiTOS2 web server. Final boundaries of structural features were decided by manually aligning the sequenced mitogenomes with previously available mitogenomes of *C. japonica* (NC003408.1, MW574361.1, KX712089.1; [8,19]) and *C. coturnix* (MW574359.1; [8]). tRNA secondary structure prediction was conducted using tRNAscan-SE2.0 and verified with the MiTOS2 results [20]. Each identified protein-coding gene (PCG) was translated using the ExPASy-Translate tool to verify the absence of internal stop codons and checked for their respective matches through NCBI-BLAST analysis [21]. The AT and GC skews were calculated using the formulas $\frac{\left(A-T\right)}{\left(A+T\right)}$ and $\frac{\left(G-C\right)}{\left(G+C\right)}$, respectively [22]. Circular genome maps of the sequenced mitogenomes were generated using Proksee (https://proksee.ca/, accessed on 17 January 2024) [23]. Codon, amino acid, and relative synonymous codon (RSCU) usage parameters of the sequenced mitogenomes were calculated using the CAIcal server (https://ppuigbo.me/programs/CAIcal/, accessed on 25 November 2023) [24].

### 2.3. Detecting Selection Pressure 

We estimated selection pressure on individual genes within the *C. japonica* and *C. coturnix* mitogenomes using the ratio of non-synonymous vs. synonymous mutations (*dN/dS*). We also searched for signatures of episodic diversification (positive selection) across the branches of a phylogenetic tree using several approaches. The *dN/dS* ratio was used to infer the type of selection on PCGs. A *dN/dS* ratio less than 1 indicates purifying (negative) selection, while a ratio greater than 1 suggests diversifying (positive) selection [25]. Using the phylogenetic trees generated in this study, we examined the following mitogenomes: *C. japonica* (PP209356), *C. coturnix* (PP212854), *Alectoris chukar* (FJ752426; [26]), *Gallus gallus* (CM028585.1), *Margaroperdix madagarensis* (MW574377.1; [8]), *Coturnix pectoralis* (MW574362; [8]), *Coturnix delegorguei* (MW574360; [8]), *Coturnix chinensis* (AB073301; [27]), *Coturnix ypsilophora* (MW574363; [8]), *Pternistis swainsonii* (MW574387; [8]), *Tetraogallus himalayensis* (KR349185; [28]), and *Tetraogallus tibetanus* (KF027439; [29]). Using the selected mitogenomes, a gene tree for each of the 13 PCGs was created in IQTREE v1.6.12 (model = GTR + I + G, bootstrap = 10,000) and used for executing selection analysis across various programs [30]. First, to calculate the *dN/dS* ratio for individual PCGs, we employed site models implemented in EasyCodeML version 1.41 [25,31]. Second, to detect positive selection along different branches of the phylogenetic tree, we analyzed each gene tree using several methods: (i) the adaptive branch-site random effects likelihood (aBSREL) and (ii) the Branch-site Unrestricted Statistical Test for Episodic Diversification (BUSTED) (both hosted on the Datamonkey 2.0 web server), as well as (iii) the branch-site models in EasyCodeML [32,33,34]. Third, to visually examine the conservation of amino acid sequences among the *Coturnix* species, we translated all PCGs and aligned them using ClustalOmega [35].

### 2.4. Genetic Distance and Phylogenetic Analyses 

In order to avoid ambiguity in sequence quality and annotation, we downloaded mitogenomes of species within Phasianidae carrying the tag ‘Reference Sequence (RefSeq)’ from the NCBI GenBank. However, all the available mitogenomes of the *Coturnix* genus (irrespective of the RefSeq tag) were also included in all distance and phylogenetic analyses (Appendix A). Genetic distance was estimated amongst the selected mitogenomes using two approaches. We calculated the genetic distance amongst each individual mitogenome to reveal the species-based sequence divergences (species n = 71). Additionally, mitogenomes were grouped into their respective genera to evaluate the genus-based sequence divergences (genus n = 31). For both analyses, we employed the Maximum Composite Likelihood Model in MEGA X and estimated genetic distances [36].

For the phylogenetic analyses, the 13 PCGs from the seventy-one mitogenomes were concatenated and used in this study. Alignment was achieved using ClustalOmega [35]. A maximum likelihood (ML) tree was constructed using the tools in IQTREE v1.6.12. To identify the optimal substitution model for our dataset, we employed ModelFinder within IQ-TREE, which selected the GTR + F + R5 model as the best fit based on the lowest Akaike Information Criterion (AIC) and Bayesian Information Criterion (BIC) scores [37]. The ML tree was then constructed and run for 10,000 iterations of ultrafast bootstrapping. Construction of the Bayesian inference (BI) tree was carried out in MrBayes 3.2.7a using the best-fit model of GTR + I + G [38]. BI tree construction was achieved using four independent chains running for 10,000,000 generations and trees being sampled every 5000 generations. During the run, the average standard deviation of the split frequencies was monitored to be under 0.01 at the end of the run, indicating convergence among the independent chains. The mixing of chains, effective sample sizes (ESS > 200), and stationarity were assessed using Tracer 1.7 [39]. The first 25% of the sampled trees were discarded as burn-in, and the remaining trees were used to build a consensus BI tree. FigTree v1.4.4 was utilized for editing the derived ML and BI phylogenetic trees, with *Columba livia* (GU908131; [40]) designated as the out-group [41].

### 2.5. Divergence Time Estimation 

In order to utilize fossil-based age constraints to calibrate the tree, we selected eight additional mitogenomes to be included for divergence dating (n = 79) (Appendix A). The fossils were selected following previously published phylogenetic reports on Galliformes [9,11,42]. Lognormal priors and fixed hard minimum ages were applied to the four fossil calibrations used in this analysis (Appendix A). A dataset consisting of the concatenated PCGs from the 79 mitogenomes was created and aligned using ClustalOmega. In order to visualize the placement of fossil priors in a phylogenetic tree, an ML tree was constructed using IQTREE (model = GTR + F + R5, bootstrap = 10,000). The placement of fossils was indicated by A, B, C, and D on the tree (Appendix A). BEAST v2.6.7 was used to carry out the divergence time estimation, and BEAUTi v2.6.7 was used to create the XML file [43]. Using BEAUTi, we selected a relaxed uncorrelated lognormal clock model, the Yule process speciation model, and the GTR + γ site model. Using BEAST, we ran Markov Chain Monte Carlo (MCMC) chains for 350 million generations that sampled trees every 10,000 generations. Initially, runs for 100 or 200 million generations were set up; however, ESS values > 200 were observed beyond 300 million generations. Tracer 1.7 was used to check convergence between runs and estimate ESS values (>200). The first 25% of the trees were discarded as burn-in, and the remaining trees were summarized using TreeAnnotator v2.6.7 [43]. FigTree v1.4.4 was used to visualize the tree, posterior probability support values, and node ages. The Bayesian analyses (BEAST and Mr. Bayes) were conducted on the Smithsonian High Performance Computing Cluster (SI/HPC), Smithsonian Institution (https://doi.org/10.25572/SIHPC, accessed on 7 December 2023).

### 2.6. Ancestral Range Estimation 

We estimated the probability of possible ancestral ranges of *Coturnix* quails using BioGeoBEARS in R [44,45]. The time-calibrated ultra-metric tree inferred from BEAST was pruned to contain seven *Coturnix* and one *Alectoris* species (as the out-group) using the R packages *ape* and *phytools* [46,47]. BioGeoBEARS allows fitting probabilistic biogeographic models with user-defined geographic areas. Initially, the terrestrial zoogeographic zones of Afro-Tropical, Madagascan, Saharo-Arabian, Palearctic, Sino-Japanese, Oriental, Oceanian, and Australian were coded as geographic areas [48]. However, the ancestral range estimations for each geographic area recovered very low probability scores. Hence, we condensed the distributions to four areas: (i) A (Oceanian and Australian), (ii) B (Oriental), (iii) C (Palearctic and Sino-Japanese), and (iv) D (Saharo-Arabian, Afro-Tropical, and Madagascan). Following species distribution ranges, we coded each species as present or absent from the assigned geographical areas [49]. Six biogeographic models were tested to identify the best-fitting model for our pruned dataset: (i) dispersal–extinction–cladogenesis (DEC) model, (ii) DEC + J, (iii) dispersal vicariance analysis (DIVA)-like model, (iv) DIVALIKE + J, (v) BayArea-like model, and (vi) BayArea-like + J built in BioGeoBEARS. The DEC model allows all variations in cladogenetic and/or anagenetic speciation events during its implementation [44,50]. The DIVA-like model allows all variations in anagenetic processes but permits only narrow sympatric cladogenetic events (i.e., does not allow daughter lineages to each inherit the ancestor’s entire range) [42,51]. Similarly, the BayArea-like model allows all variations in anagenetic processes but permits only widespread sympatric cladogenetic events (i.e., does not allow daughter lineages to inherit portions of their ancestor’s range) [42,52]. Incorporation of the ‘+J’ parameter, indicative of founder-event speciation, was also evaluated across all three models. This parameter represents a rare dispersal event that instantaneously leads to the formation of a geographically isolated new lineage, derived from one or a few individuals [44]. In this study, the BayArea-like + J model was selected as the best-fitting model using the weighted AIC, and ancestral ranges of *Coturnix* species were estimated accordingly.

We also conducted an analysis where we reclassified the western Palaearctic and eastern Palearctic/Sino-Japanese regions as two separate ancestral areas. Consequently, we conducted a modified BioGeoBEARS run with reclassified geographic areas as follows: (i) A (Oceanian and Australian), (ii) B (Oriental), (iii) C (western Palaearctic), (iv) D (Saharo-Arabian, Afro-Tropical, and Madagascan), and (v) E (eastern Palearctic and Sino-Japanese). All six biogeographic models were tested, and DEC + J was identified as the best-fitting model for our modified run. The inferences drawn from this modified run were compared with the range reconstruction patterns obtained with the BayArea-like + J model from the previous analysis.

## 3. Results

### 3.1. Gene Arrangement, Nucleotide Composition, and Codon Usage Analysis

Sequencing yielded ~100 million paired-end reads for both *C. coturnix* and *C. japonica*. FastQC analysis revealed that the base call quality of all raw reads was within the range denoted as ‘very good quality calls’. The average Phred quality score across all reads exceeded Q30 for both species and was deemed suitable for mitogenome assembly. Complete mitogenomes of *C. japonica* and *C. coturnix* were assembled, annotated, and submitted to NCBI GenBank under the accession codes PP209356 and PP212854, respectively. The depth of the mapped mitogenome reads was estimated to be 48× for *C. japonica* and 107× for *C. coturnix*. Despite sequencing a nearly identical number of reads for both species, observed variations in the depth of mitochondrial genomes retrieved suggest disparities in the distribution of library fragments across the genome. These disparities may stem from minute differences in reagents/template handling, or the quality of the input DNA. The length of the sequenced mitogenomes was calculated at 16,698 bp (*C. japonica*) and 16,696 bp (*C. coturnix*). The lengths of previously published *C. japonica* mitogenomes (NC003408.1, MW574361.1, and KX712089.1) are 16,697 bp, 16,701 bp, and 16,698 bp, respectively, and *C. coturnix* (MW574359.1) is 16,700 bp. Both mitogenomes sequenced in this study are composed of 13 PCGs, 22 transfer RNAs (tRNA), 2 ribosomal RNAs (rRNA), and a mitochondrial control region (CR) (Figure 1). Amongst the structural features, eight tRNAs (trnaQ, trnA, trnN, trnC, trnY, trnS2, trnP, and trnE) and one PCG (*nad6*) were located on the light chain, while the remaining features were situated on the heavy chain (Appendix A). The nucleotide composition for various structural features of *C. japonica* and *C. coturnix* mitogenomes was calculated and described (Appendix A). Overall, both mitogenomes displayed similar base compositions across features. PCGs, tRNAs, and rRNAs displayed higher A-T (55.6% and 55.7%) content relative to G-C (44.4% and 44.3%) content. The control region of both mitogenomes was calculated to contain negative A-T and G-C skews, but overall, the A-T content of the control region was estimated to be highest amongst the structural features (59.8% and 60.7%).

The tRNA secondary structures of *C. japonica* and *C. coturnix* mitogenomes displayed a clover leaf structure for all tRNAs except tRNA(Serine), which contained a mismatch base pair in its dihydrouridine arm (Appendix A). For both sequenced mitogenomes, ‘ATG’ was the start codon for all PCGs except for the *cox1* gene (GTG was the start codon). Both mitogenomes were found to have incomplete stop codons in *cox3* and *nad4*.

The new *C. coturnix* mitogenome (PP212854) generated from a specimen collected in India was compared for sequence variation against a previously available but geographically undefined mitogenome of *C. coturnix* (MW574359) [8]. We identified one gap and five differences in nucleotides across the two mitogenomes, estimating a sequence similarity score of 99.8%. Similarly, we compared the new *C. japonica* mitogenome (PP209356) for sequence variation against previously available *C. japonica* mitogenomes (KXY712089, MW57436, and NC003408). There was ~100% sequence similarity amongst the *C. japonica* mitogenomes, without any gaps. Furthermore, *C. japonica_*PP209356 was closest to *C. japonica_*KXY712089 in sequence similarity (~100%). Overall, sequence dissimilarity amongst *C. coturnix* mitogenomes was found to be slightly more pronounced than the *C. japonica* sequences investigated in this study.

Codon usage analysis for *C. japonica* and *C. coturnix* mitogenomes sequenced in this study revealed identical preferences in amino acid usage and RSCU patterns. Both *C. japonica* and *C. coturnix* mitogenomes were biased towards codons such as TTA, TTG, CTT, CTC, CTA, and CTG and amino acids such as leucine, serine, threonine, and isoleucine (Appendix A). RSCU analysis for both mitogenomes revealed the highest preferences in codons such as TCC, TCA, TCG, AGT, and AGC and amino acids such as leucine, serine, proline, threonine, and alanine (Appendix A).

### 3.2. Selection Pressure

The *dN/dS* ratio of all PCGs was found to be less than 1, indicating neutral/purifying selection on all PCGs (Figure 2). Amongst the PCGs, *cytb* and *cox1* were the most conserved genes, whereas *atp8* was estimated to be the least conserved. Furthermore, the *dN/dS* ratio of all PCGs was almost identical for both *C. japonica* and *C. coturnix*, suggesting they were subjected to very similar evolutionary pressures.

Each of the 13 PCG gene trees was tested, and except for the *nad4* gene, investigations across all other gene trees (across all branches) found no evidence of episodic diversifying selection in our phylogenies. EasyCodeML, aBSREL, and BUSTED detected highly significant evidence of episodic diversifying selection across the *Coturnix* lineage for the *nad4* gene tree (Figure 2). Our findings suggest that for the *nad4* gene tree, the *Coturnix* lineage experienced adaptive evolution, wherein non-synonymous substitutions were favored in response to changing environmental pressure or ecological niches. Visual inspection of the *nad4* amino acid sequence alignment for the *Coturnix* species shows a large number of variable amino acid sites across species (Appendix A). Our investigation identified 44 putative sites of non-synonymous substitutions within the *Coturnix* lineage. Amongst these, the most frequent substitutions were alanine to threonine (n = 5), followed by threonine to isoleucine (n = 4), and methionine to leucine (n = 4). The analysis of non-synonymous substitution sites showed that amino acids such as threonine (n = 15), isoleucine (n = 11), leucine (n = 11), methionine (n = 10), valine (n = 9), serine (n = 9), and alanine (n = 7) were most prevalent.

### 3.3. Phylogenetic Analyses

Pairwise genetic distances calculated for the sequences used in this study revealed *C. japonica* and *C. coturnix* to be the closest to each other, with a pairwise genetic distance of 2.1%. Furthermore, distances revealed that *C*. *delegorguei* and *C*. *pectoralis* are closest to *C. japonica* and *C. coturnix,* followed by *M. madagarensis*, *C*. *ypsilophora,* and *C*. *chinensis,* respectively (Appendix A). Similarly, genus-based comparisons revealed the *Tetraogallus* and *Pternistis* genera are closest to *Coturnix* (Appendix A).

Phylogenetic analyses revealed similar topologies for the ML and BI trees (Figure 3). The bootstrap values (ML tree) and posterior probabilities (BI tree) for the phylogenetic tree were 100 and 1, respectively, for 68 out of 71 branches, signifying high support for these branches. Additionally, proper mixing of chains and ESS values > 1500 provided evidence for a high-quality BI analysis (Appendix A). The phylogenetic analysis reveals that the ‘*Coturnix*’ clade bifurcates into distinct lineages: one comprising *C. chinensis* and *C. ypsilophora*, and the other encompassing the remaining species. Within the latter group, *M. madagarensis* diverges as a separate lineage, while *C. coturnix, C. japonica, C. delegorguei,* and *C. pectoralis* form a closely related cluster. Subsequent branching within this cluster further delineates the relationships, with *C. coturnix* and *C. japonica* demonstrating a particularly close phylogenetic affinity.

### 3.4. Divergence Time Estimation

Using a relaxed molecular clock, the maximum clade credibility tree supported a mid-Miocene divergence of *Coturnix* clade members, around 13.89 Ma (HPD; highest posterior density: 10.65–17.23) (Figure 4 and Appendix A). The divergence between *C. chinensis* and *C. ypsilophora* was estimated at 8.59 Ma (HPD: 5.13–12.17). The divergence between *M. madagarensis* and *C. coturnix, C. japonica, C. delegorguei,* and *C. pectoralis* was estimated at 10.09 Ma (HPD: 7.10–13.10). The branches of *C. coturnix* and *C. japonica* and *C. delegorguei* and *C. pectoralis* diverged at 7.2 Ma (HPD: 4.77–9.80). While *C. delegorguei* and *C. pectoralis* diverged around 4.13 Ma (HPD: 2.15–6.34), the divergence between *C. coturnix* and *C. japonica* is the youngest within the *Coturnix* clade at 2.25 Ma (HPD: 1.18–3.57).

### 3.5. Ancestral Range Reconstruction of Coturnix

Our biogeographic analysis determined that the BayArea-like + J model was the optimal fit for our time-calibrated ultrametric tree. In our study, the *Coturnix* clade, as one of the only Phasianidae genera capable of long-distance dispersal, is well represented by this model choice (Figure 5).

The most recent common ancestor (MRCA) of the *Coturnix* clade is inferred to have spanned the Oriental and Australian (including Oceanian) zones. Ancestors of *C. chinensis* and *C. ypsilophora* dispersed from the Oriental zone towards the Australian landmass. The remaining members of *Coturnix* appear to have dispersed from the Oriental zone into the Palearctic, Sino-Japanese, Saharo-Arabian, Afro-Tropical, and Madagascan zones. Our results also indicate the MRCAs of *M. madagarensis* and *C. coturnix, C. japonica, C. delegorguei,* and *C. pectoralis* were extant across the Oriental, Palearctic, Sino-Japanese, Saharo-Arabian, Afro-Tropical, and Madagascan zones. Subsequently, *C. delegorguei* and *C. pectoralis* might have dispersed from the Oriental zone to colonize the African and Australian landmasses, respectively. Similarly, our biogeographic reconstructions for *M. madagarensis* likely suggest dispersal from the Oriental to the Afro-Tropical and Madagascan zones and its subsequent endemism on Madagascar. Our findings suggest the ancestral populations of *C. coturnix* and *C. japonica* were distributed across the Oriental, Palearctic, and Sino-Japanese zones. Furthermore, we hypothesize that *C. coturnix* dispersed to Eurasia and Africa (central and western Palearctic, Saharo-Arabian, Afro-Tropical, and Madagascan zones), while *C. japonica’s* dispersion was primarily directed towards eastern Asia (eastern areas of Palearctic and Sino-Japanese zones) (Appendix A). Ancestral range estimation carried out in which the western Palaearctic and eastern Palearctic/Sino-Japanese regions were considered two separate ancestral areas, with the DEC + J model determined to be the best-fitting, also elucidated the origin of *C. coturnix* and *C. japonica* ancestors in the Oriental zone, with subsequent dispersal towards Eurasia/Africa and eastern Asia, respectively (Appendix A).

## 4. Discussion

### 4.1. Nucleotide Composition, Codon Usage, and Selection Analysis

High A-T content in the control region of the sequenced mitogenomes is predicted to aid in transcriptional and translational regulation of PCGs [53]. The arrangement of genes in and around the control region of *C. japonica* and *C. coturnix* resembles the ancestral avian gene order [54,55]. The structural features, gene arrangements, control region gene order, and A-T indices of the sequenced mitogenomes were conserved across the members of Phasianidae, including in the previously reported *Coturnix* mitogenomes [26,27,56]. These structural features are also highly conserved across vertebrate taxa (including Aves) [15,18,57,58]. Homogeneity in the patterns of codon, amino acid, and RSCU usage in the mitogenomes of *C. japonica* and *C. coturnix* suggests identical translational efficiency of PCGs and highly similar evolutionary pressures acting on them [59].

We used site models in CodeML (through EasyCodeML) to identify the *dN/dS* ratio of each PCG in the *C. japonica* and *C. coturnix* mitogenomes [60,61]. Vertebrate mitogenomes tend to remain highly conserved across speciation events [62]. Consequently, prior research on avian mitogenomes has inferred little selection pressure on individual PCGs, and our findings are consistent with these observed patterns [15,18,63,64]. Over the past two decades, multiple approaches have been developed to assess selection pressure across a phylogeny. However, the choice of algorithms is known to affect the test of selection in these approaches [65]. Hence, we used three algorithms (EasyCodeML, aBSREL, and BUSTED) to assess selection pressure across our limited *Coturnix* phylogeny. The branch-site model in EasyCodeML requires assignment of a foreground lineage (in this study, the *Coturnix* clade), whereas aBSREL and BUSTED require no prior branch assignment. All three algorithms detected highly significant evidence of episodic diversifying selection across the *Coturnix* lineage for the *nad4* gene tree.

Previous reports on bird mitogenomes revealed that nad4 showed evidence of positive selection in *Eopsaltria australis*, *Pygoscelis papua*, and high-altitude Galliformes (*Tetraophasis szechenyii*, *Tetraophasis obscurus*, *Lophurus ilhuysii*, *Crossoptilon crossoptilon*, *Perdix hodgsoniae*, and *Tetraogallus tibetanus*) [29,66,67]. In *E. australis* and *P. papua*, positive selection was attributed to environmental adaptation or variability, whereas in high-altitude Galliformes, this selection is proposed to aid in high-altitude acclimatization. We similarly hypothesize that the *nad4* gene of the Coturnix clade experienced positive selection pressure due to environmental changes and acclimatization requirements during its evolutionary history. Furthermore, our results of positive selection, supported by multiple algorithms (CodeML, aBSREL, and BUSTED), might reduce the risk of identifying false-positive signals [66]. Despite these precautions, we recognize that interpreting evolutionary signals from single mitogenomes may lead to overestimation [68]. For example, a study on primate mitogenomes presented various hypotheses for positive selection within mitochondrial genes without attributing them to a single definitive cause [65]. Nevertheless, the signs of positive selection detected within the Coturnix clade appear to be robust, and the implications of these findings may be explored in future studies.

### 4.2. Phylogenetics and Divergence Dating

To minimize ambiguity in nomenclature, we have adhered to the species names listed in the NCBI GenBank Taxonomy database. Specifically, the species *Coturnix chinensis* and *Coturnix ypsilophora* are the same as *Synoicus chinensis* and *Synoicus ypsilophorus*. These species are included under the term ‘members of the *Coturnix* clade’ throughout this study. Additionally, the generated phylogenetic trees highlight that *C. coturnix, C. japonica, C. delegorguei,* and *C. pectoralis* form a monophyletic clade for the genus *Coturnix* and support the replacement of *C. chinensis* and *C. ypsilophora* into the genera *Excalfactoria* or *Synoicus* [69,70]. Our phylogenetic tree mirrors the one generated in Kimball et al. [8] and is similar to the topologies of *Coturnix* species reported in Kimball et al. [7] and Stein et al. [10]. Moreover, our findings are also consistent with prior phylogenies derived from complete mitochondrial genomes, although these studies contained a smaller number of taxa, ranging from 22 to 39 species [26,40,71].

Our divergence dating results align closely with those reported in earlier studies by Stein et al. [10], Wang et al. [11], and Chen et al. [9], underscoring the significance of mitochondrial genome-based phylogenies, particularly when extensive resources for generating large autosomal genomic datasets are not available. The limited discrepancies observed in the divergence dates may arise from the exclusive use of mitogenomes and the limited sampling of Galliformes, including the omission of certain ‘key’ species, the inclusion of which was beyond the scope of our study [11]. Therefore, our divergence time estimation and ancestral range reconstruction discussions primarily focus on the *Coturnix* clade, with special attention to the relationship between *C. coturnix* and *C. japonica*.

The Miocene epoch was characterized by a warmer global climate and the emergence of kelp forests and grasslands [72]. These conditions may have provided favorable conditions for *Coturnix* quails to expand their range and diversify [1]. Previous researchers reconstructing the biome occupation history of Galliformes estimated the divergence of the Phasianidae occurred during the middle to late Miocene, lending credence to our results [73]. Additionally, the expansion of temperate broadleaf deciduous forests during the mid-Miocene epoch is hypothesized to have played a crucial role in facilitating the diversification and subsequent range expansion of Phasianidae, including species within the *Coturnix* genus [73]. Within the Phasianidae family, the divergence of *C. coturnix* and *C. japonica* represents one of the most recent diversification events within the family, with only the genus *Chrysolophus* showing a more recent divergence at 1.68 Ma (HPD: 0.73–2.83) (Figure 4). The diversification of *C. coturnix* and *C. japonica* is estimated to have taken place during the Pleistocene epoch of the Quaternary period, a time marked by repeated expansions and contractions [74]. Additionally, the Pleistocene was characterized by glacial periods during which ice sheets could have served as barriers or land bridges, potentially influencing the biogeographic distribution of *C. coturnix* and *C. japonica* [75,76].

### 4.3. Biogeography

The best-fitting BayArea-like + J model incorporates anagenetic changes and most cladogenetic processes, with the exception of subset-sympatric speciation and vicariance, while also accounting for founder-event (+J) speciation through long-distance dispersal [44,77]. This model has been effectively applied to explain the biogeographic patterns of numerous continental clades across different taxa [44,77].

Inferring from reconstructed ancestral ranges, we postulate that *C. chinensis* and *C. ypsilophora* dispersed from the Oriental zone to the Australian landmass, aided by lower sea levels and the exposure of land bridges [11,78]. The estimated divergence of *C. chinensis* and *C. ypsilophora* at approximately 8.59 Ma suggests that the possible timeframe for their long-distance dispersal events from the Oriental zone to the Australian landmass likely occurred during the late Miocene period. The initiation of bidirectional avifaunal dispersal between the Oriental zone and the Australian landmass is estimated to have commenced around the end of the Oligocene, approximately 23 Ma [78,79]. Significant dispersal of Galliformes from the Oriental zones to the Australian landmass is posited to have occurred during the later stages of the Miocene and into the Pliocene [79,80].

Overall, the findings in this study are consistent with the previous reports by Wang et al. [11] and Chen et al. [81], which documented the ancestral distribution of Phasianidae in the Oriental zone and subsequent dispersal to Africa, Eurasia, Australia, and North America. Furthermore, the divergence dates estimated within the *Coturnix* clade in this study, along with those reported in prior studies, suggest that global vicariance events, such as the breakup of Gondwana, far preceded the diversification of *Coturnix* on the geological time scale [11,82]. Instead, the biogeographic history of these species is more likely explained by range expansions and dispersal events across major landmasses. The dispersal events between the Oriental and Afro-Tropical (including Madagascan) zones have previously been reported to have taken place in both directions [11,83]. We infer that *M. madagarensis* dispersed from the Oriental to the Afro-Tropical and Madagascan zones by transient land bridges rather than stochastic marine events, and such a hypothesis is supported by Hosner et al. [82]. Furthermore, the recent work by Masters et al. [84] reported three transient land bridges connecting Africa and Madagascar at approximately 66–60 Ma, 36–30 Ma, and 12–05 Ma, which may have facilitated the dispersal events that contributed to Madagascar’s rich biodiversity. Considering the divergence time estimates for *M. madagarensis* in our study (10.09 Ma; HPD: 7.10–13.10) and its dispersal pattern, it is plausible that *M. madagarensis* reached Madagascar via one of these land bridges, particularly the one that was present between 12–05 Ma.

Our findings also suggest *C. delegorguei* and *C. pectoralis* might have undertaken dispersal events from the Oriental zone to colonize the African and Australian landmasses, respectively. The range reconstruction for *C. delegorguei* and *C. pectoralis* in this study reveals a pattern of disjunct distribution, reminiscent of the distribution of ratites in Australia and Africa [85].

In light of our findings and inferences from previous research, we postulate that the origins of both *C. coturnix* and *C. japonica* ancestors can be traced back to the Oriental zone [11]. Previous studies have described a pattern of bidirectional dispersal among phasianids, including movements from the Oriental to African regions, between Africa and the Palearctic, and between the Indo-Malaya/Sino-Japanese and Australian/Oceanian regions [82,83]. The ancestral range reconstructions of *C. coturnix* and *C. japonica* in our study support previously established dispersal trajectories yet reveal distinct dispersal routes for each species. In summarizing our analyses on *C. coturnix* and *C. japonica* mitogenomes, we identified: (i) similarity in gene arrangement and nucleotide composition; (ii) identical translational mechanisms (codon and amino acid usage) and selection pressure; (iii) a high degree of phylogenetic relatedness within phasianids with a relatively recent divergence; and (iv) a shared biogeographic origin with subsequent divergent dispersal patterns leading to distinct geographic distributions (Appendix A). Additionally, the previous literature indicates near identical plumage coloration, natural history, habitat types, and breeding patterns for *C. coturnix* and *C. japonica* [1,3,4,49]. In light of such evidence, we propose that interspecific interference and competitive exclusion may have played critical roles in shaping the evolutionary paths and current geographic distributions of *C. coturnix* and *C. japonica*. Recent discussions on avian evolution have highlighted the role of competition, particularly interspecific territoriality, where birds of different species vie for territory [86,87]. Previous reports suggest that interspecific territoriality is more likely to occur among closely related and recently diverged species (<5 Ma), where there is a significant overlap in ecological niches and potential for hybridization [86,87]. Particularly, Drury et al. [86] discuss patristic distance (genetic closeness) and plumage dissimilarity as key predictors of territorial disputes among closely related bird species. Interspecific territoriality is perceived as a maladaptive by-product of confusion in territorial signals (e.g., plumage, mates, and ecological niches), and the genetic basis for these signals diminishes over time due to natural selection, leading species to evolve distinct traits to minimize competition [86,87]. In the case of *C. coturnix* and *C. japonica*, their indistinguishable plumage, genetic closeness, and relatively recent common ancestry might facilitate and exacerbate interspecific territoriality. Hence, despite their common origin, the species may have adopted divergent dispersal patterns to avoid overlap and confusion in territorial signals (i.e., avoid interspecific territoriality), leading to distinct geographic distributions. Adoption of such a strategy by *C. coturnix* and *C. japonica* could have possibly evolved to circumvent possible competition for resources, mates, or habitats.

Furthermore, we propose that the dispersal routes adopted by *C. coturnix* and *C. japonica* were significantly influenced by the Tibetan Plateau. The Tibetan Plateau forms a unique geological barrier, which has affected the distribution and evolution of biodiversity in and around the region [88,89]. It has played a crucial role in enabling the intercontinental movement of various species, including substantive biotic transfer between the Indian subcontinent and Eurasia, since ~70 Ma [90]. In the context of birds, the Tibetan Plateau orchestrates the convergence of migratory paths due to its extensive geography, compelling birds to navigate around its western and eastern flanks [89,91]. Adopting divergent dispersal routes around the plateau has been identified as a reproductive barrier, leading to reduced hybridization rates and enhanced genetic differentiation among bird populations [89,91]. Consequently, the current geographic distributions of *C. coturnix* and *C. japonica* are the outcomes of their distinct dispersal strategies, primarily devised to evade interspecific territoriality and shaped by the Tibetan Plateau’s geographic constraints. Although in this study we adhered to the widely accepted convention that *C. coturnix* and *C. japonica* represent two different species despite their documented ability to hybridize in field and laboratory settings, it would be compelling to explore broader patterns of territoriality and competition between them in areas of habitat overlap and hybridized populations such as Mongolia and northeastern India [3,92,93].

## 5. Conclusions

Our study provides valuable insights into *C. coturnix* and *C. japonica* gene arrangement, nucleotide composition, codon usage, and selection pressures, revealing a high degree of structural conservation and similarity in translational mechanisms. Our findings align with previous studies, indicating that despite similar life history traits and genetic relatedness, *C. coturnix* and *C. japonica* have evolved distinct dispersal routes and occupy unique geographical ranges. Furthermore, the divergence time estimates suggest that *Coturnix* diversified during the mid-Miocene, and their current distribution appears to be a result of dispersal rather than global vicariance events. Our results also suggest that to minimize overlap and reduce interspecific competition, *C. coturnix* and *C. japonica* adopted divergent dispersal routes influenced by the geographic barrier of the Tibetan Plateau. Future research may focus on the ecological and evolutionary mechanisms underlying these dispersal patterns. Although our study is limited by the absence of extensive genomic sampling of the nuclear genome, it significantly advances our understanding of *Coturnix* biogeography and evolution.

## Figures and Tables

**Figure 1 genes-15-00742-f001:**
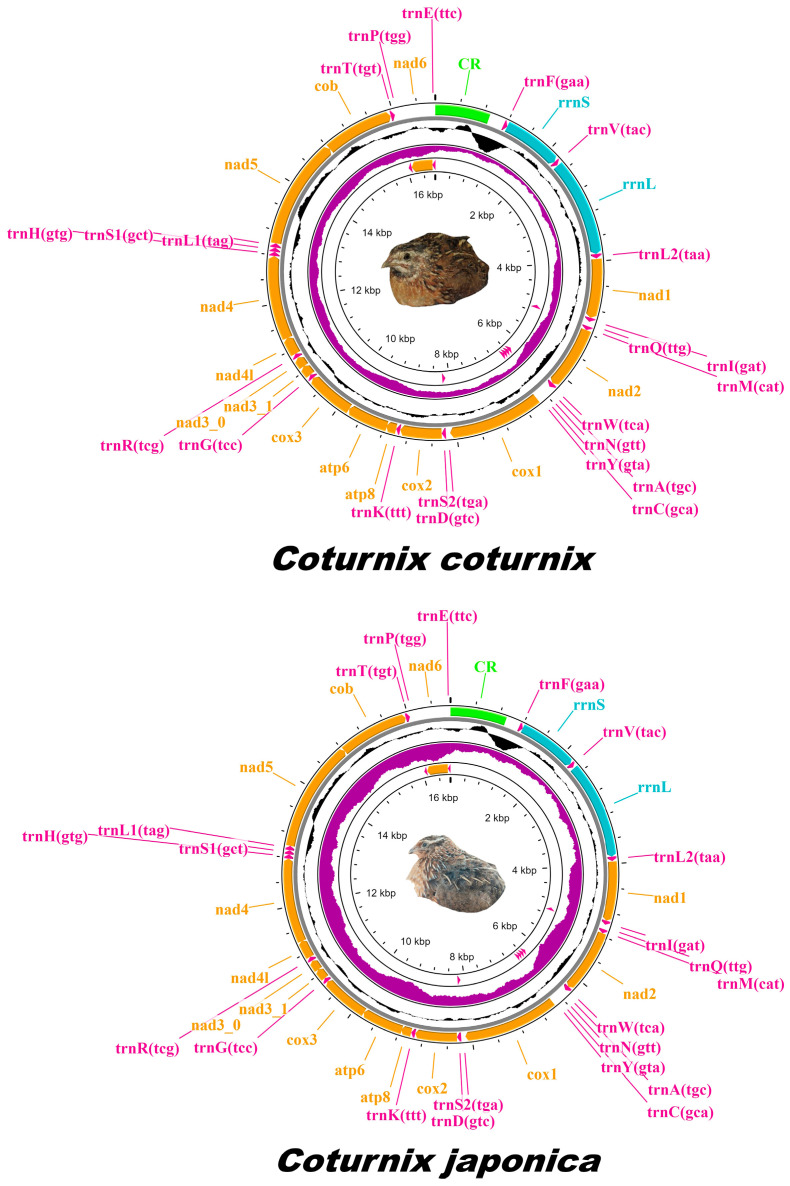
Circular schematic map of the mitogenomes of *C. coturnix* and *C. japonica*. Genes are represented with different color blocks, with PCGs in orange, rRNAs in blue, tRNAs in magenta, and the control region in green. Black sliding windows correspond to the GC content, and purple sliding windows correspond to the value of the GC skew. Letters in parentheses for tRNA tags indicate anticodons. The arrow direction at the end of protein-coding genes indicates transcription on the plus (right) or minus (left) strand.

**Figure 2 genes-15-00742-f002:**
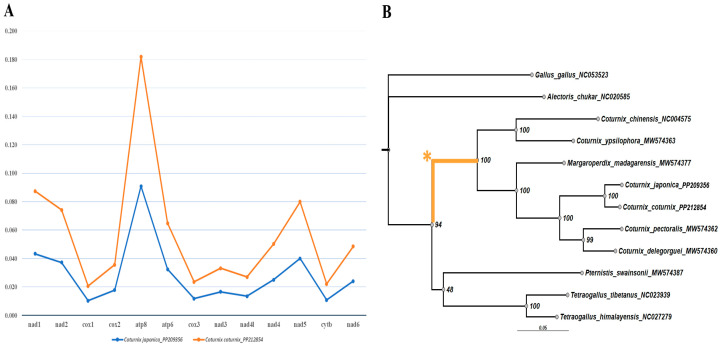
(**A**) *dN/dS* estimates for 13 protein-coding genes of *C. coturnix* and *C. japonica* mitogenomes sequenced in this study. The X-axis shows the gene name, and the Y-axis shows the units of the *dN/dS* ratio. (**B**) Maximum likelihood tree constructed using the *nad4* gene of selected mitogenomes used in this study. EasyCodeML, aBSREL, and BUSTED algorithms detected positive selection across the highlighted tree branch that leads to the *Coturnix* lineage (boldly marked in yellow ochre and asterisked).

**Figure 3 genes-15-00742-f003:**
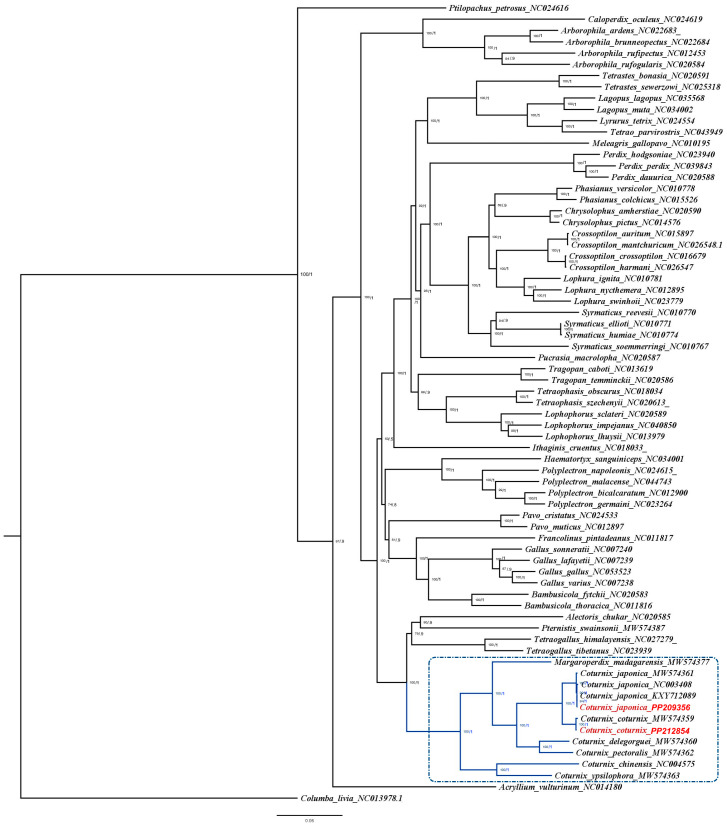
Topology of maximum likelihood (ML) and Bayesian inference (BI) trees generated for the 71 Phasianidae mitogenomes used in this study. *C. livia* is used as the out-group to root the trees. The numbers at each node represent bootstrap values for ML analyses (0–100%) and posterior probabilities for BI analyses (0–1). Branches of *Coturnix* clade members are in blue and highlighted in the dotted dark blue box. The *C. coturnix* and *C. japonica* mitogenomes sequenced in this study are highlighted in red text.

**Figure 4 genes-15-00742-f004:**
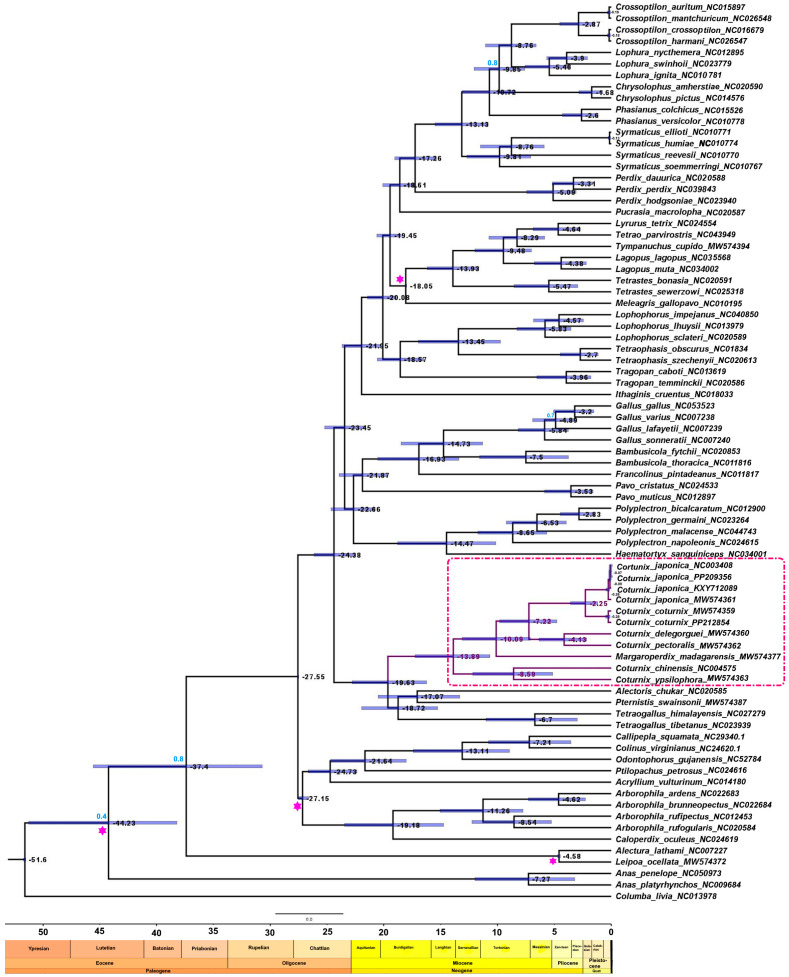
BEAST maximum clade credibility tree showing divergence time estimates among Phasianidae species based on four fossil calibrations and rooted with the out-group, *C. livia*. Posterior probabilities are indicated only at nodes for values < 1.0 in light blue. Mean divergence time estimates are shown in black next to the nodes, and purple bars show the lower and upper bounds of the 95% highest posterior density (95% HPD) interval for the time estimates. The branches of the *Coturnix* clade members are in red and highlighted in a dotted red box. The crimson asterisk indicates the nodes supported by fossil calibrations.

**Figure 5 genes-15-00742-f005:**
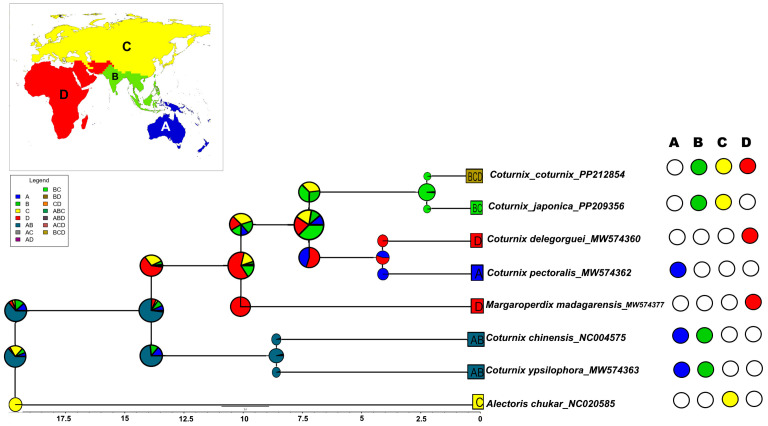
The ancestral biogeographic area reconstruction estimated using the BayArea-like + J model generated in BioGeoBEARS. The BEAST maximum clade credibility tree was pruned to contain seven *Coturnix* and one *Alectoris* species (as the out-group). The estimated ancestral areas are colored according to the four zoogeographic zones: (i) A (Oceanian and Australian), (ii) B (Oriental), (iii) C (Palearctic and Sino-Japanese), and (iv) D (Saharo-Arabian, Afro-Tropical, and Madagascan). The color palette on the nodes of the chronogram shows the combination of most likely ancestral areas estimated by BioGeoBEARS. The coding scheme for each geographic area for each species, present or absent, is shown on the right of the tree. Insert: biogeographic areas used in the ancestral area reconstruction.

## Data Availability

The associated BioProject, BioSample, and SRA numbers for *C. coturnix* whole genome reads are PRJNA1069095, SAMN39610750, and SRR27727407, respectively. The associated BioProject, BioSample, and SRA numbers for *C. japonica* whole genome reads are PRJNA840867, SAMN28561720, and SRR19344531, respectively. Mitogenomes assembled from *C. coturnix* and *C. japonica* whole genome reads are stored under accession codes PP212854 and PP209356, respectively.

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
