# Peer review of "Mitogenomic Insights into the Evolution, Divergence Time, and Ancestral Ranges of Coturnix Quails"

_genes, 2024, doi:10.3390/genes15060742_

Round 1
Reviewer 1 Report
Comments and Suggestions for Authors
The manuscript reports the determination of two complete mitogenomes from the Old World quail genus Coturni. These data are combined with GenBank data for other mitogenomes from the genus and from other species, mostly from the family, Phasianidae, to which Coturnix belongs. The combined data, or subsets of them, were analysed to investigate questions such as the mitochondrial genome nucleotide composition, gene order, whether there is evidence for selection, the divergence times of Coturnix species (plus their close relatives) and the ancestral distribution of Coturnix taxa. Complete mitogenomes are already available for the species from which they were collected here, so the main novelty of the work is the analytical use that is made of the data. Whilst I appreciate the amount of effort that has been made in the analyses, I have major reservations about how well they support the conclusions that are drawn from them, particularly those regarding biogeographical questions.
Some of these reservations derive from uncertainty about the taxonomic status of the species from which the new mitogenomes were sequenced. These are designated as two distinct species Coturnix coturnix and Coturnix japonica. This designation is controversial, and the taxa might better be considered subspecies as they hybridise in both laboratory and field conditions, including in the overlap of their recent native ranges (see the Barilani et al. and Chazara et al. References listed below). Such hybridisation, which is not mentioned in the manuscript, raises very serious questions about the interpretation of the ancestral range reconstruction. The most obvious scenario to test is that such divergence as there is between the taxa predominantly occurred allopatrically in a previously wider ranging species which had become geographically divided into for example southern and eastern Asian regions or between the eastern and western Palaearctic . Testing this would require at a minimum that the eastern Palearctic/Sino-Japanese region be treated as an “ancestral area” separately to the western Palaearctic. The Materials and Methods section (lines 217–223) indicates that the reconstruction is very dependent on the areas that are assumed in the analysis, with low probability scores being found for a seven-area analysis. If similarly low scores are found for an analysis in which the eastern Palaearctic/Sino-Japanese region is treated separately then it should be recognised that the dataset is not suitable for answering the question of the ancestral ranges of C. coturnix and C. japonica.
The discussion of dispersal on lines 525–564 is apparently predicated on some type of sympatric speciation (“a shared biogeographic origin” on line. 525). Athough this is not made explicit, it is an assumption of the selected model in BioGeoBears. It is difficult to envisage a plausible model of “widespread sympatric cladogenesis” that could accommodate the known hybridisation between C. coturnix and C. japonica. Some of the material in these paragraphs, especially on the biogeographic effects of the Tibetan Plateau, could be adapted to other parts of a revised discussion that are not limited to a particular model.
Whether dispersal is short or long range is not established in the manuscript. Although long-range dispersal is frequently mentioned, so also are land-bridges that would facilitate short-range dispersal. It would be better to omit reference to the type of dispersal or to treat specified modes as tentative assumptions.
I have made annotated on the attached PDF. These include grammatical corrections or suggestions, indications of the need for more information so that analyses could be repeated and deletions of material that does not add much meaning to the text either because it is repetitious or simply states common understanding. In some cases, such material is indicated to be deleted, without comment.
The figures are generally well presented. However, it would be helpful to the reader to indicate nodes supported by fossils in Figure 4.
I would prefer the Results and Discussion sections to be separate, Partly this is because the “results and Discussion” section of the present manuscript is so long. More importantly, however, the separation might enable the authors to delimit with more clarity, those parts of the Discussion that are firmly based on the present results and those that are more speculative.
References:
Barilani, M., Derégnaucourt, S., Gallego, S., Galli, L., Mucci, N., Piombo, R., ... & Randi, E. (2005). Detecting hybridization in wild (Coturnix c. coturnix) and domesticated (Coturnix c. japonica) quail populations. Biological Conservation, 126(4), 445-455.
Chazara, O., Minvielle, F., Roux, D., Bed’hom, B., Feve, K., Coville, J. L., ... & Rognon, X. (2010). Evidence for introgressive hybridization of wild common quail (Coturnix coturnix) by domesticated Japanese quail (Coturnix japonica) in France. Conservation Genetics, 11, 1051-1062.

Language use is generally good. I have made a few corrections and suggestions on the attached PDF.
Author Response
Response to Assistant Editor
We would like to thank Dr. Lia Yan (Assistant editor) and the Reviewers for their constructive remarks, which were of great help to improve this manuscript. We have revised the manuscript in track-change mode incorporating the changes suggested by the Reviewers. The revised manuscript is submitted for further consideration along with a point-by-point rebuttal letter for each of the query raised. Further, we have re-checked the manuscript for any English language errors and fluency in text is assured by the corresponding author being a native English speaker.
Reviewer #1
Comment 1: Complete mitogenomes are already available for the species from which they were collected here, so the main novelty of the work is the analytical use that is made of the data.
Response: We are grateful to the reviewers for their thorough reading of the manuscript. We would like to emphasize that Coturnix coturnix has a widespread distribution across Asia, Europe, and Africa, and is suspected to comprise many perceived subspecies (McGowan et al. 2023). In the absence of comprehensive studies in this direction, the first mitogenome of C. coturnix from India, generated in this study, will be crucial for future research aimed at elucidating the subspecies categories of C. coturnix.
Reference:
- McGowan, P. J. K., G. M. Kirwan, E. de Juana, and P. F. D. Boesman (2023). Common Quail (Coturnix coturnix), version 1.1. In Birds of the World (S. M. Billerman, Editor). Cornell Lab of Ornithology, Ithaca, NY, USA. https://doi.org/10.2173/bow.comqua1.01.1
Comment 2: Whilst I appreciate the amount of effort that has been made in the analyses, I have major reservations about how well they support the conclusions that are drawn from them, particularly those regarding biogeographical questions. Some of these reservations derive from uncertainty about the taxonomic status of the species from which the new mitogenomes were sequenced. These are designated as two distinct species Coturnix coturnix and Coturnix japonica. This designation is controversial, and the taxa might better be considered subspecies as they hybridise in both laboratory and field conditions, including in the overlap of their recent native ranges (see the Barilani et al. and Chazara et al. References listed below). Such hybridisation, which is not mentioned in the manuscript, raises very serious questions about the interpretation of the ancestral range reconstruction.
Response: We understand the reviewer’s reservation about the uncertainty of the taxonomic status of Coturnix coturnix and Coturnix japonica, as C. japonica has previously been recognised as a race of C. coturnix (Madge and McGowan 2002). Following McGowan et al. (2023), we adhered to the widely accepted convention of adopting Coturnix coturnix and Coturnix japonica as two different species. C. coturnix and C. japonica exhibit distinct vocalizations, breeding plumage and significant genetic divergence. Additionally, they have been documented as sympatric in Mongolia and in northeastern India. Therefore, we categorized C. coturnix and C. japonica as two separate species for this study. However, we believe the information regarding hybridization and their potential for inter-breeding is important for the study. Hence, we have incorporated it in the ‘Discussion’ section of the manuscript and cited the references as mentioned by the reviewer (line no. 844-849).
Reference:
- Madge, S., and P. McGowan (2002). Pheasants, Partridges and Grouse, including Buttonquails, Sandgrouse and Allies. Christopher Helm, London, UK.
- McGowan, P. J. K., G. M. Kirwan, E. de Juana, and P. F. D. Boesman (2023). Common Quail (Coturnix coturnix), version 1.1. In Birds of the World (S. M. Billerman, Editor). Cornell Lab of Ornithology, Ithaca, NY, USA. https://doi.org/10.2173/bow.comqua1.01.1
Comment 3: The most obvious scenario to test is that such divergence as there is between the taxa predominantly occurred allopatrically in a previously wider ranging species which had become geographically divided into for example southern and eastern Asian regions or between the eastern and western Palaearctic. Testing this would require at a minimum that the eastern Palearctic/Sino-Japanese region be treated as an “ancestral area” separately to the western Palaearctic. The Materials and Methods section (lines 217–223) indicates that the reconstruction is very dependent on the areas that are assumed in the analysis, with low probability scores being found for a seven area analysis. If similarly low scores are found for an analysis in which the eastern Palaearctic/Sino-Japanese region is treated separately then it should be recognised that the dataset is not suitable for answering the question of the ancestral ranges of C. coturnix and C. japonica.
Response: Following reviewer’s suggestion, we re-ran BioGeoBEARS with modified ancestral areas. We coded the ancestral areas as (i) A (Oceanian and Australian), (ii) B (Oriental), (iii) western Palaearctic (C), (iv) eastern Palearctic and Sino-Japanese (E) and (v) D (Saharo-Arabian, Afro-Tropical and Madagascan). All 6 models were tested and the DEC+J model was determined to be the best fit for the data set. The reconstructed ancestral ranges (using 5 ancestral areas) (Figure S11) did exhibit lower confidence for reconstructed areas compared to the previous analysis using four areas (Figure 5). However, the most recent common ancestor (MRCA) of C. coturnix and C. japonica showed a possible presence across the Oriental zoogeographic region, followed by subsequent dispersal in two different directions. C. japonica exhibited dispersal towards eastern China and Japan, whereas C. coturnix was estimated to have dispersed over the Indian sub-continent, Africa and Europe. The results using modified parameters strengthens our original findings regarding C. coturnix and C. japonica biogeography. We have incorporated this new finding in our manuscript (line no. 243; 526).
While we believe that complete mitogenomes are a valuable tool for biogeographic reconstructions, we also highlight the caveat of relying solely on mitogenomic evidence for such analyses in the “Discussion” section of our manuscript.
Comment 4: The discussion of dispersal on lines 525–564 is apparently predicated on some type of sympatric speciation (“a shared biogeographic origin” on line. 525). Athough this is not made explicit, it is an assumption of the selected model in BioGeoBears. It is difficult to envisage a plausible model of “widespread sympatric cladogenesis” that could accommodate the known hybridisation between C. coturnix and C. japonica. Some of the material in these paragraphs, especially on the biogeographic effects of the Tibetan Plateau, could be adapted to other parts of a revised discussion that are not limited to a particular model.
Response: We acknowledge your reservation regarding the potential impact of a preconceived model choice on the outcome of our BioGeoBEARS inference. We also understand that the best fit model identified in this study might not fully accommodate the evolutionary scenarios involving hybridization between C. coturnix and C. japonica populations.
However, this study is aimed at utilizing Coturnix mitogenomes to understand its biogeography and evolution, and emphasizing on specific instances of hybridization might be outside the purview of the study. Additionally, Matzke (2013) emphasizes that there is no "true" model for biogeographic range reconstruction in BioGeoBEARS, while encouraging to deduce inferences after testing all possible scenarios of cladogenetic and anagenetic events through the six models. In line with this, our inferences and discussions are primarily derived from the ancestral range reconstructed using the BayArealike+J model, which assumes widespread sympatric cladogenesis. However, we also considered the analyses done by other models in the same analyses. All the models returned a similar ancestral range reconstruction pattern for C. coturnix and C. japonica albeit with varying degrees of confidence.
Following your advice, we conducted a modified BioGeoBEARS run using five ancestral areas with the best fit model choice of DEC+J. This analysis returned the same biogeographic pattern, thereby strengthening the outputs of this study. We have revised the manuscript to integrate the material on the biogeographic effects of the Tibetan Plateau into a separate “Discussion” and we have emphasized that our results are not limited to a particular model.
Reference:
- Matzke, Nicholas J. (2013). Probabilistic historical biogeography: new models for founder-event speciation, imperfect detection, and fossils allow improved accuracy and model-testing. Frontiers of Biogeography, 5(4), 242-248. http://escholarship.org/uc/item/44j7n141
Comment 5: Whether dispersal is short or long range is not established in the manuscript. Although long-range dispersal is frequently mentioned, so also are land-bridges that would facilitate short-range dispersal. It would be better to omit reference to the type of dispersal or to treat specified modes as tentative assumptions.
Response: We agree with you and recognize the tentative nature of the assumptions made. Consequently, we have revised the text to remove any instances of “long-range dispersal”.
Comment 6: I have made annotated on the attached PDF. These include grammatical corrections or suggestions, indications of the need for more information so that analyses could be repeated and deletions of material that does not add much meaning to the text either because it is repetitious or simply states common understanding. In some cases, such material is indicated to be deleted, without comment.
Response: We are thankful for the detailed edits. We have incorporated all the edits as advised.
Response to some of the questions/comments are as follows:
“How are the gaps treated”: MEGA X uses default settings to estimate gaps in sequence alignments. Response: No special input was needed.
“This reads as if concatenation were done before alignment, which would not be appropriate.”: Response: Phrases edited
“How was this determined? Response: Line 184”: GTR+I+G model was nearest to the best fit models suggested by ModelFinder, that is incorporated in MrBayes 3.2.7a software, hence was used in this study.
“for what parameters, line 206” Response: For all parameters estimated by Tracer software.
“Why are bases written in lower case in this figure. Line 272” Response: Default setting within Proksee tool used. To adjust spacing, we insist to let the bases written in lower case.
“To minimize…...this study, line 373-377” Response: We believe this information fits better into Discussion, hence we have moved it there.
“Meaning, line 381” Response: It means the chains should attain stationarity during the Bayesian runs.
“Relevance to quails? line 433” Response: No direct relevance, Kelps mentioned as both Kelps and Grasslands emerged during the mentioned time frame.
Comment 7: The figures are generally well presented. However, it would be helpful to the reader to indicate nodes supported by fossils in Figure 4.
Response: We have edited the figure to highlight nodes supported by fossils in Figure 4.
Comment 8: I would prefer the Results and Discussion sections to be separate, Partly this is because the “results and Discussion” section of the present manuscript is so long. More importantly, however, the separation might enable the authors to delimit with more clarity, those parts of the Discussion that are firmly based on the present results and those that are more speculative.
Response: We have separated the Results and Discussion into different sections for better clarity.
Reviewer 2 Report
Comments and Suggestions for Authors
The authors elucidated the evolutionary trajectory and ancestral distribution patterns through a thorough analysis of the mitochondrial genomes for Coturnix coturnix and Coturnix japonica. The results advanced the understanding of the biogeographic and evolutionary processes leading to the diversification of C. coturnix and C. japonica, laying important groundwork for further research on this genus.
The authors have done a very good job and put a lot of effort into gaining new knowledge about these two species.
Page 2 Line 87-89:
→Various tissues (including testes, identifying the sample as male) from the carcass were sampled and stored 88 in DESS buffer.
“Various tissues”, what kind of various tissues? Were all of the various tissues used for DNA extraction?
Page3 Line 102:
→For C. japonica, a commercially farmed specimen from India was sampled……
“commercially farmed specimen” not “wild species”, does this affect the biogeographic and evolutionary analysis results?
Page 10:
For Figure 3, there is no “_” in the species name in the tips, please delete. And the NCBI accession numbers should be regular font.
The same for Figure 4 and Figure 5.
Author Response
Reviewer #2
Comment 1: Page 2 Line 87-89: Various tissues (including testes, identifying the sample as male) from the carcass were sampled and stored 88 in DESS buffer. “Various tissues”, what kind of various tissues? Were all of the various tissues used for DNA extraction?
Response: We are thankful for the encouraging review of the manuscript. We have described the various tissues that were sampled and the exact tissue used for DNA extraction the manuscript (line no. 88).
Comment 2: Page3 Line 102: A For C. japonica, a commercially farmed specimen from India was sampled…… “commercially farmed specimen” not “wild species”, does this affect the biogeographic and evolutionary analysis results?
Response: The sequence divergence scores was estimated to be nearly 100% for all the C. japonica mitogenomes used in this study. With such genetic closeness we are confident that the biogeographic and evolutionary analysis results are not impacted by the usage of farmed specimen of C. japonica.
Comment 3: Page 10: For Figure 3, there is no “_” in the species name in the tips, please delete. And the NCBI accession numbers should be regular font. The same for Figure 4 and Figure 5.
Response: We apologize to the Reviewer for the appearance of “_” in the species name, however due to script requirements and usage of multiple softwares for generating images, we kindly insist on keeping the format of names/NCBI accession numbers names as such to maintain uniformity.
Round 2
Reviewer 1 Report
Comments and Suggestions for Authors
I consider that the authors have successfully addressed the comments I made on the original version of the manuscript. I have only a few minor suggestions on this revision (referring to line numbers)
Reverse italic font for “Coturninx” in title on line 462
499: “suggest likely” rather than “likely suggests” (noting that “reconstructions” is plural).
508: replace “elucidated” by a different word - “postulated” is possible but doesn’t seem quite tight.
Maybe include an introductory sentence at the start of the Discussion. E.g. The newly collected complete mitogennomes from Coturnix illuminate the evolution and biogeography of the genus by ...” etc.
646: Insert “range” or “population” before “expansions”
674: maybe “long” instead of “far”